# UNRAVELING HALLUCINATION IN LARGE REASONING MODELS: A TOPOLOGICAL PERSPECTIVE

## ABSTRACT

Large Reasoning Models (LRMs) have recently demonstrated strong capabilities in multi-step problem solving through extended chain-of-thought (long-CoT) and self-reflective reasoning. However, the very reliance on long reasoning chains makes them vulnerable to hallucinations, where early-stage errors become amplified and embedded within otherwise coherent logical traces. Existing hallucination detection methods largely focus on short-CoT models, leaving the unique challenges of LRMs underexplored. In this paper, we propose a topological perspective to *analyze*, *detect*, and *mitigate* hallucinations in LRMs. **(I) Analyze**: We formalize reasoning trajectories as structured graphs and conduct statistical analysis on 6,000+ annotated reasoning graphs, revealing 17 topological features that reliably distinguish hallucinated from faithful reasoning. **(II) Detect**: Building on these insights, we develop `G-Detector`, a graph-based post-hoc hallucination detector that leverages only reasoning topology and achieves up to $88.9\%$ detection accuracy. **(III) Mitigate**: We extend `G-Detector` to mitigation by filtering high-risk reasoning traces during cold-start supervised fine-tuning in the LRM training process, which improves the LRM's factual accuracy by $13.8\%$ without impairing reasoning ability. Studies showcase that hallucinations in LRMs are not arbitrary but leave identifiable structural signatures in their reasoning topologies, opening a principled pathway toward reliable detection and prevention of LRM hallucinations. Codes are available at https://anonymous.4open.science/r/GDetector.

## 1 INTRODUCTION

Large Reasoning Models (LRMs) (Hou et al., 2025; OpenAI, 2025; Jaech et al., 2024; Qwen, 2025; Yu et al., 2025) have recently attracted substantial attention for their capacity to conduct multi-step reasoning via structured Chain-of-Thought (CoT) (Wei et al., 2022; Zhang et al., 2023) and self-reflection mechanisms. The development paradigm of LRMs typically follows a two-stage pipeline: supervised fine-tuning (SFT) for cold-start initialization, followed by reinforcement learning (RL) to incentivize long-horizon reasoning. This paradigm has given rise to several landmark LRMs, including the `OpenAI o1/o3/o4` (Jaech et al., 2024), `DeepSeek R1` (Guo et al., 2025a), `Kimi K2` (Team et al., 2025) and `Qwen3` (Yang et al., 2025).

Paradoxically, the very long-CoT training paradigm that underpins the success of LRMs has also raised serious concerns regarding their reliability, particularly with respect to hallucination. Extended reasoning chains are prone to amplifying early-stage errors, as initial inaccuracies can be iteratively revised, elaborated, or reframed throughout the reasoning process (Lu et al., 2025). Such hallucinations become deeply embedded within logically coherent traces, rendering incorrect content more persuasive and substantially harder to detect. Prior detection methods have primarily focused on **standard CoT hallucination**, employing strategies such as external knowledge verification (Min et al., 2023; Bayat et al., 2023), self-checking algorithms (Manakul et al., 2023), or supervision over hidden states (Farquhar et al., 2024). However, these approaches typically target non-thinking models that rely solely on CoT prompting, applied to relatively simple tasks with short reasoning chains. In contrast, **hallucination detection for LRMs** remains underexplored: while recent attempts exist (Sun et al., 2025), they rely heavily on white-box access to model internals and remain confined to narrow knowledge domains. More critically, they fail to capture and exploit the distinctive long-CoT dynamics of LRMs for hallucination modeling.

**Our Work.** Given the deficiencies, we hypothesize that mitigating LRM hallucinations fundamentally requires leveraging their internal reasoning dynamics. Long-CoT reasoning chains inherently incorporate self-reflection processes, such as self-loops, alongside more complex structures, including knowledge clustering (cliques) and parallel reasoning paths. These previously overlooked topological cues provide critical signals for *analyzing*, *detecting*, and *mitigating* hallucinations. Motivated by this insight, we formulate the following three central questions that guide our study:

> 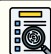 (**Analyzing Hallucination**) *Do LRMs' reasoning chains harbor recognizable, generalizable, and easily accessible signatures of hallucination?*

In Section 3, we introduce the *topology of reasoning*, which formalizes the computational states, their interconnections, and the terminal conclusion paths of LRM reasoning chains into structured reasoning graphs. By constructing and annotating over 6,000 such graphs, we conduct rigorous $t$-tests on topological features of hallucinated versus non-hallucinated subsets (*e.g.*, self-loops, diameter, clique count). This analysis reveals 17 statistically significant distinctions, establishing that hallucinated and non-hallucinated reasoning paths inherently exhibit structural differences.

> 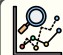 (**Detecting Hallucination**) *Can these topological signatures function as reliable indicators for hallucination detection?*

Having established these topological distinctions, a natural next step is to examine their practical utility. In Section 4, we develop `G-Detector`, a swift hallucination detector for LRMs that relies solely on reasoning graph topology, achieving up to $88.9\%$ detection accuracy without external databases, additional LLM supervision, or multi-sampling, and outperforming state-of-the-art baselines like FactTool and CPP by $17.31\%$.

> 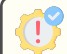 (**Mitigating Hallucination**) *Can these insights offer a principled pathway for mitigating hallucinations of LRMs from the origin?*

In Section 5, we harness `G-Detector` to mitigate LRM hallucinations at their origin. By filtering high-risk hallucinated reasoning examples detected by `G-Detector` during the cold-start SFT phase of LRM training, we steer the model away from incorporating topologically possible hallucination patterns. Empirical evaluation shows that this filtration improves the factual accuracy of the resulting LRM by up to $14.8\%$ on SimpleQA and TriviaQA, while preserving general reasoning capabilities.

Overall, our topologically grounded study introduces a new paradigm for understanding, detecting, and mitigating hallucinations in LRMs. By leveraging solely the intrinsic reasoning structures within long-CoT outputs, we can inherently identify traces of hallucination. `G-Detector` not only enables precise detection but also lays the foundation for addressing hallucinations at their origin, offering a principled pathway toward more reliable and faithful LRM reasoning.

## 2 PRELIMINARY & BACKGROUND

**Large Reasoning Models (LRMs).** LRMs are inspired by the chain-of-thought (CoT) paradigm: decomposing complex problems into intermediate sub-goals and allocating additional tokens to "think" before producing a final answer (Wei et al., 2022; Zhang et al., 2023). Following early prompt-based CoT approaches, which equipped LLMs with step-by-step reasoning through few-shot exemplars, more recent work increasingly employs RL to explicitly reinforce reasoning behaviors (Guo et al., 2025b; He et al., 2025). Prior to RL fine-tuning, it is common to leverage a stronger LRM to generate long-CoT traces for cold-start SFT, adapting models to the reasoning format. This *SFT+RL* strategy has emerged as a prevailing paradigm across domains, such as deep research (Zheng et al., 2025; Nguyen et al., 2025), and tool-integrated reasoning (Qian et al., 2025).

**Hallucination in LRM.** Despite the substantial gains LRMs achieve from CoT-style step-by-step reasoning, they are simultaneously more prone to hallucinations, including logical inconsistencies, factual fabrications, and contextual contradictions (Yao et al., 2025). Detecting such errors can, in principle, rely on conventional hallucination detection techniques for LLMs, such as cross-referencing with external knowledge bases (Chern et al., 2023; Sansford et al., 2024), self-consistency checks (Manakul et al., 2023; Xue et al., 2025), or probing hidden states (Kossen et al., 2024; Liao et al., 2025; Sriramanan et al., 2024). However, as highlighted by Lu et al. (2025), these methods often yield limited effectiveness and suffer from severe inefficiency (sometimes requiring

days of computation). To address hallucinations specific to the long-CoT dynamics of LRMs, we introduce a specialized topological view to analyze and detect LRM hallucinations.

**Reasoning Topology.** The *topology of reasoning* has been widely used to characterize and visualize the complex states of long-CoT. Early designs were relatively simple, typically adopting chain-like (Wei et al., 2022; Wan et al., 2024; Wang et al., 2025b; Yu et al., 2023) or tree-like structures (Fu et al., 2023; Wu et al., 2025; Wang et al., 2025a). However, as LLM reasoning capabilities advance, emergent self-reflective phenomena render reasoning chains far from linear, introducing numerous loops and backtracking structures. Graphs, by virtue of their structural expressiveness, have thus become a more comprehensive choice. Graph-of-Thought (Besta et al., 2024a; Yao et al., 2024) and Forest-of-Thought (Bi et al., 2025) explicitly model LLM reasoning paths as graphs, where nodes correspond to reasoning states and edges represent transitions between states. A recent study by Google DeepMind (Minegishi et al., 2025) systematically demonstrates how to transform an LRM's CoT chain into a *reasoning graph*, which serves as a foundational inspiration for our work.

## 3 ANALYSIS: DECIPHERING HALLUCINATION GRAPHS

In this section, we first describe the process of translating a long CoT from an LRM into a structured *reasoning graph* (▷ Section 3.1), collect four distinct subsets of reasoning graphs (▷ Section 3.2), and identify 17 graph-theoretic features exhibiting significant differences between hallucinated and non-hallucinated graphs (▷ Section 3.3), thereby substantiating the existence of inherent structural properties underlying hallucination phenomena.

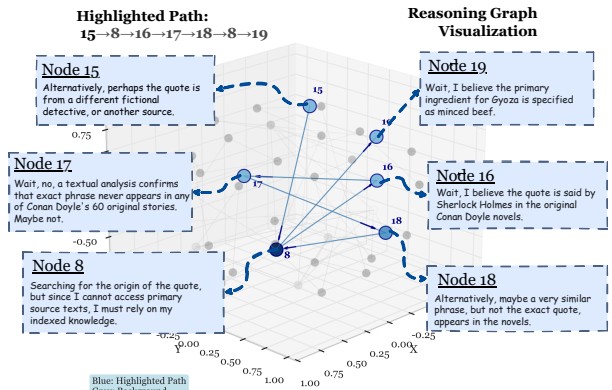

Figure 1: A 3D visualization of a reasoning graph, highlighting the self-correction path. The path from node 8 to 18 represents the exploration of an incorrect hypothesis, which, upon invalidation, triggers a strategic rollback to the pivotal node (8) for a new reasoning path.

| Category | Node Text |
|---|---|
| Recalling | ➤ Okay, what the Rosetta Stone is? ➤ I need to remember what Emperor Penguins are. ➤ Okay, so I need to figure out which king signed the Magna Carta. |
| Providing Examples | ➤ For example, the Battle of Hastings was fought in 1066. ➤ For example, The Great Gatsby is by F. Scott Fitzgerald, published in 1925. |
| Self Reflection | ➤ Wait, I think the correct year is 1911. ➤ Let me think again. Maybe I'm mixing up their capitals. ➤ But I need to confirm. |

Table 1: Representative node texts from reasoning graphs, classified by their primary cognitive function: information recall, example provision, and metacognitive self-reflection.

### 3.1 TRANSLATING CoT CHAINS INTO REASONING GRAPHS

To enable a topological investigation of hallucinations, we formalize the mapping of an LRM's long-CoT into a structured *reasoning graph*. Let $\mathcal{Q} = \{q_n\}_{n=1}^N$ denote the set of $N$ evaluation questions. For each $q \in \mathcal{Q}$, the model produces a sequence of intermediate reasoning segments $\mathcal{R}_q = (r_1, r_2, \ldots, r_{T_q})$, where each $r_i$ corresponds to a contiguous block of tokens representing a distinct reasoning step, and $T_q$ is the number of segments for $q$. Each segment $r_i$ is encoded as a continuous vector by averaging the hidden states across its tokens at transformer layer $\ell$, yielding the set of all segment embeddings across $\mathcal{Q}$:

$$s_i^{(\ell)} = \frac{1}{|r_i|} \sum_{t=1}^{|r_i|} h_{i,t}^{(\ell)}, \quad h_{i,t}^{(\ell)} \in \mathbb{R}^d, \quad \mathcal{S} = \bigcup_{q \in \mathcal{Q}} \{s_1^{(\ell)}, \ldots, s_{T_q}^{(\ell)}\}, \tag{1}$$

where $h_{i,t}^{(\ell)}$ denotes the hidden state of token $t$ in segment $r_i$, and $\mathcal{S}$ aggregates all segment representations for subsequent clustering and reasoning graph construction.

**Node & Edge Construction.** Following the graph construction process in Minegishi et al. (2025), we cluster $\mathcal{S}$ using $K$-means to produce centroids $\{c_k\}_{k=1}^K$. Each centroid $c_k$ defines a node $v_k$ in the reasoning graph, resulting in $\mathcal{V} = \{v_1, \ldots, v_K\}$. Each segment $r_i$ is assigned to its nearest

centroid $i^* = \arg\min_k \|s_i^{(\ell)} - c_k\|_2$ (calculated by the Euclidean distance in embdding space), and consecutive segments are linked to form a directed edge, yielding the reasoning graph $\mathcal{G}_q$:

$$\mathcal{G}_q = (\mathcal{V}, \mathcal{E}_q), \quad \mathcal{E}_q = \{(v_{i_j} \to v_{i_{j+1}}) \mid j = 1, \ldots, T_q - 1\},$$

which preserves both the sequential flow of the LRM's reasoning and latent structural dependencies. Figure 1 illustrates a constructed reasoning graph, where the path "15 $\to$ 8 $\to$ 16 $\to$ 17 $\to$ 18 $\to$ 8 $\to$ 19" exemplifies a classical self-correction process in LRMs: the model initially follows a trajectory from Node 8, but upon reaching Node 17 recognizes an inconsistency ("Wait, no, ...") and consequently returns to Node 8 to initiate a revised reasoning path. Table 1 displays several functional roles of different reasoning nodes within reasoning graphs. Overall, the reasoning graph effectively captures the intrinsic structure underlying long-CoT reasoning dynamics.

## 3.2 CURATING REASONING GRAPHS

**Long CoT Curation & Annotation.** Our initial step involves the systematic generation and annotation of long CoT corpora. We employ a diverse suite of LRMs, denoted as $\mathcal{M} = \{$ `Qwen3-8B`, `Qwen3-14B`, `DeepSeek-Distill-Qwen-7B`, `DeepSeek-Distill-Qwen-32B` $\}$, to produce reasoning chains. For each factual question $q_i$ from a source dataset $\mathcal{Q}_{\text{src}}$, a reasoning chain $\mathcal{R}_{m,i}$ is generated by a model $m \in \mathcal{M}$. Each chain is subsequently subjected to an annotation function, $\Lambda : \mathcal{R} \to \{0, 1\}$, which assigns a binary hallucination label $y_i = \Lambda(\mathcal{R}_{m,i})$, where $y_i = 1$ indicates the presence of hallucination as determined by a rigorous consistency-checking protocol (see Appendix B). To probe model robustness, we define a perturbation operator $\Psi$, which synthesizes a corrupted question $q_i' = \Psi(q_i)$ by embedding a set of pre-defined false or out-of-domain facts into the original prompt $q_i$ (the specification of $\Psi$ is in Appendix B). These perturbed questions are then used to generate the perturbed set $\{\mathcal{R}_{m,i}'\}$, which are similarly annotated to yield labels $\{y_i'\}$.

**Reasoning Graph Curation.** Following the annotation process, we apply the graph transformation operator $\mathcal{T}$, as defined in Section 3.1, to map each reasoning chain $\mathcal{R}$ to its corresponding structured reasoning graph $\mathcal{G} = \mathcal{T}(\mathcal{R})$. This procedure yields a comprehensive graph corpus, which we stratify into four distinct sets based on the input's factual integrity and the output's fidelity:

- **Factual-Accurate (FA) set** $\mathcal{C}_{\text{FA}}$, where factual inputs lead to accurate outputs ($y_i = 0$);
- **Factual-Hallucination (FH) set** $\mathcal{C}_{\text{FH}}$, where factual inputs lead to hallucinated graphs ($y_i = 1$);
- **Perturbed-Accurate (PA) set** $\mathcal{C}_{\text{PA}}$, where exemplifies model resilience ($y_i' = 0$);
- **Perturbed-Hallucination (PH) set** $\mathcal{C}_{\text{PH}}$, where external misinformation causes hallucination.

Jointly examining (1) hallucination outcomes alongside (2) the presence or absence of input perturbations is helpful for disentangling two distinct underlying patterns: hallucinations induced by external interference (*external hallucinations*) and those autonomously generated by the LRM itself (*internal hallucinations*) (see findings in Section 3.3). The data statistics are summarized in Table 2.

Table 2: **Basic statistics** for the curated reasoning graph corpora, devided by the input's factual integrity (**factual** vs. **perturbed**) and the output's annotated accuracy (**accurate** vs. **hallucination**).

| Statistic | Factual-Accurate (FA set) | Factual-Hallucination (FH set) | Perturbed-Accurate (PA set) | Perturbed-Hallucination (PH set) |
|---|---|---|---|---|
| Hallucination? | ✗ | ✓ | ✗ | ✓ |
| Input Perturbation? | ✓ | ✓ | ✗ | ✗ |
| Number of Graphs | 2,000 | 1,830 | 4,50 | 2,029 |
| Avg. CoT Length (tokens) | 1828.73 | 2410.66 | 2056.81 | 1979.05 |

## 3.3 STATISTICAL INSPECTION

To characterize the structural features of hallucinatory and non-hallucinatory long-CoTs, we examine the reasoning graphs through five major categories of graph-theoretic properties (in Table 3). The overall **Scale & Breadth** of the reasoning graph is quantified by the number of nodes/edges and the average path length. **Structural Coherence** is assessed via the size of maximal cliques and average degree, as densely interconnected subgraphs are posited to represent stable, self-consistent knowledge clusters. Concurrently, **Thought Process Separation** evaluates the capacity for maintaining parallel, non-overlapping reasoning threads through the size of the maximum independent set, while degree assortativity reveals the organizational logic of interconnections. The **Path Complexity** of transitions is measured by path/hop length norms to distinguish substantive logical leaps from incremental, token-level progressions. Finally, **Cyclic Verification** is quantified through loop count and transitivity, which reflects the self-reflection or self-correction behaviour in LRM reasoning.

Table 3: Graph-Theoretic properties and their correlates in long-CoT traits.

| Category | Metric | Formula | Reasoning Correlates |
|---|---|---|---|
| Network Scale & Exploration Breadth | Number of Nodes
Number of Edges
Diameter
Average Path Length | $\|\mathcal{V}\|$
$\|\mathcal{E}\|$
$\max_{u,v \in \mathcal{V}} d(u,v)$
$\frac{\sum_{u \neq v} d(u,v)}{\|\mathcal{V}\|(\|\mathcal{V}\|-1)}$ | Cognitive scope, reasoning breadth, exploration breadth |
| Knowledge Structure Complexity | Number of Maximal Cliques
Maximum Clique Size
Ramsey Number (Clique)
Avg. Degree Connectivity | $\omega(\mathcal{G})$
$R(k,l)$
$\frac{1}{N_k}\sum_{i:k_i=k}\frac{1}{k_i}\sum_{j \in N(i)}k_j$ | Knowledge organization, conceptual coherence, logic soundness |
| Thought Process Separation | Max Independent Set Size
Ramsey Number (Ind. Set)
Degree Assortativity | $\alpha(\mathcal{G})$
$R(\alpha,\beta)$
$\frac{\sum_i (j_i - \bar{j})(k_i - \bar{k})}{\sqrt{\sum_i (j_i - \bar{j})^2}\sqrt{\sum_i (k_i - \bar{k})^2}}$ | Divergent thinking, parallel reasoning threads |
| Conceptual Leaps & Path Complexity | Metric Closure Avg Weight
Path Length Norm
Average Hop Length | $\frac{2}{n(n-1)}\sum_{1 \leq i < j \leq n} d(i,j)$
$\left(\sum_{e \in P}|w_e|^p\right)^{1/p}$
$\frac{2}{n(n-1)}\sum_{1 \leq i < j \leq n} d(i,j)$ | Logical leaps, reasoning step complexity |
| Local Structure & Cyclic Verification | Average Clustering Coeff.
Transitivity
Loop Count | $\frac{1}{\|\mathcal{V}\|}\sum_{v \in \mathcal{V}} C_v$
$(3 \times N_\triangle)/N_3$
$\nu(G) = m - n + c$ | Self-reflection, verification, iterative refinement/correction |

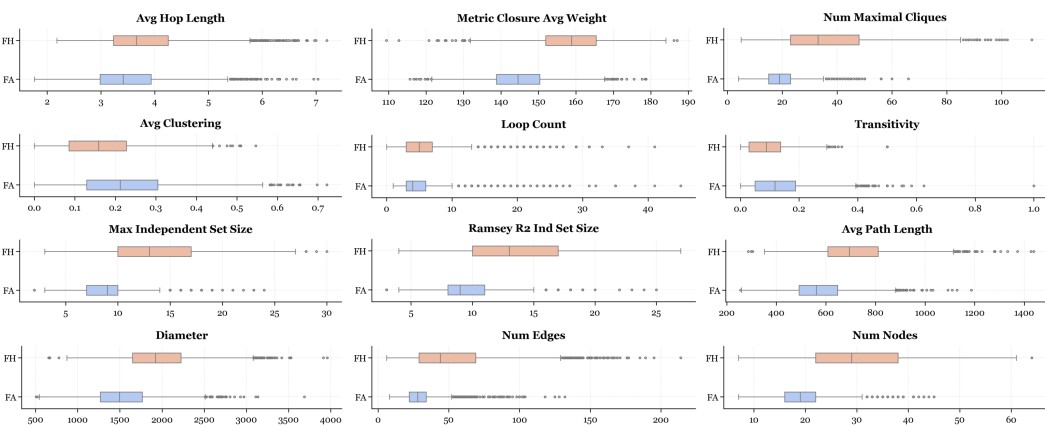

Figure 2: The topology statistical difference between FH and FA graphs.

**Key Findings.** Figures 2 and 5 highlight the most discriminative topological metrics between FH/FA and PH/PA graphs, while Figures 10 and 11 present comparisons across all four graph categories. Complementary pairwise statistical analyses of graph features (*e.g.*, $t$-values, $p$-values, and Cohen's $d$) are reported in Tables 5 to 8. Representative visualizations of graphs from each category are shown in Figure 4. From this comprehensive body of data, we distill three key findings.

> **Obs. I. Over-Complication Trap: Hallucination Arises From Over-complexity, Not Simplicity.**

Figure 2 and Table 5 reveal that internal hallucination in LRMs is not a symptom of shallow reasoning but rather a dive into an "over-complication trap." This contradicts the simplistic view of traditional LLM hallucination as mere knowledge gaps (Agrawal et al., 2024; Huang et al., 2025a). When reasoning from factual inputs, FH graphs exhibit a profound and statistically significant structural bloat compared to their FA counterparts. They are vastly larger in scale, featuring $57\%$ more nodes (30.59 vs. 19.46) and $81\%$ more edges (53.70 vs. 29.72). This indicates that the LRM, in its long-CoT process, activates an excessive number of concepts and explores a convoluted web of connections. The reasoning process itself is more tortuous

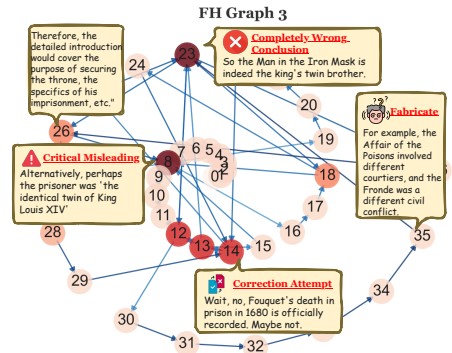

Figure 3: A case study reveals the over-complication trap in FH graphs.

and inefficient, evidenced by a $28\%$ larger graph diameter and a $25\%$ longer average path length. Crucially, this expansion signifies structural deficiency, not richness. Figure 3 illustrates how an LRM becomes entrapped in hallucination through excessive reasoning complexity. Specifically,

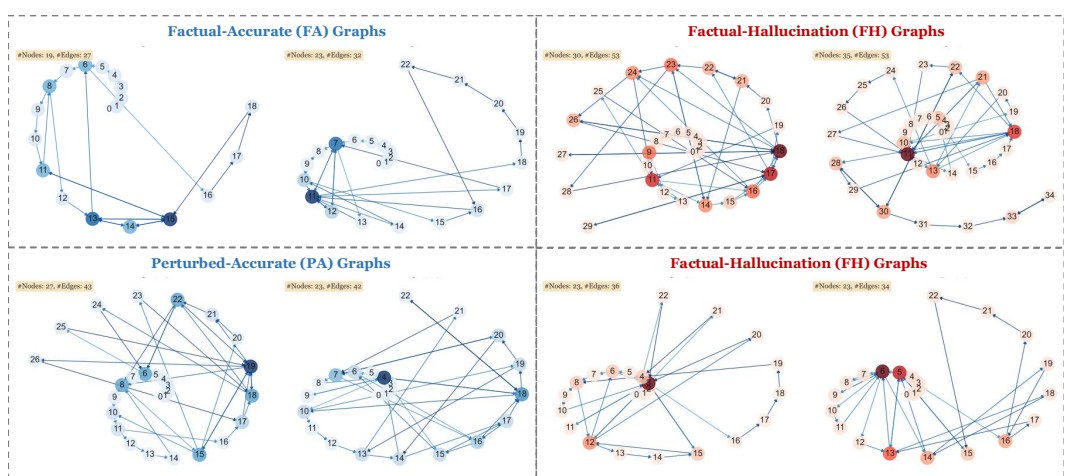

Figure 4: Visualization of reasoning graphs from four groups (FA, FH, PA, FH).

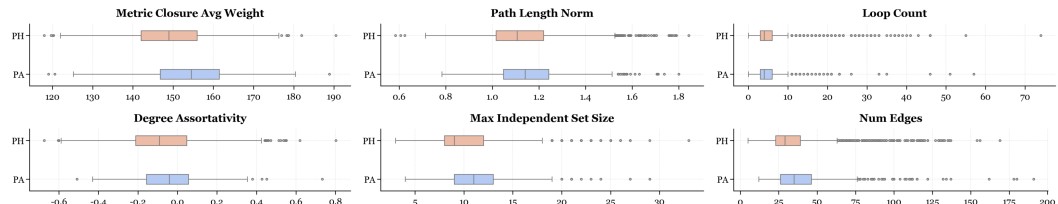

Figure 5: The topology statistical difference between PH and PA graphs.

at Node 8 the model introduces a persuasive yet fictitious premise, which it repeatedly revisits in subsequent steps (Nodes 15, 26). It then fabricates additional supporting details (Node 35) to reinforce this erroneous belief. This recursive reinforcement ultimately entrenches the initial mistake, culminating in a confidently articulated but entirely incorrect conclusion (Node 23).

**Excessive Loops.** The over-complication trap is likewise reflected in the *loop count*. Although cyclic reasoning is often regarded as a hallmark of self-verification (Minegishi et al., 2025), analysis reveals that hallucinations are distinguished by an overabundance of *inefficient* loops. Specifically, FH graphs contain significantly more loops than FA graphs (5.96 vs. 5.04, $p < 0.001$ in Figure 4), indicating unproductive cyclic thinking. PA graphs also employ a higher loop count than FA (5.69 vs. 5.04), suggesting these cycles are productive for error correction. In contrast, both internal (FH) and external (PH) hallucinatory states exhibit a similarly high number of loops, which, unlike in the PA case, fail to guide the reasoning to a correct conclusion. This suggests that while successful reasoning uses loops for verification, hallucinated reasoning gets trapped in these recursive patterns.

> **Obs. II. Resilience to Misinformation Requires Broader Reasoning Exploration.**

Comparing FA and PA reasoning graphs reveals the significant *topological cost of resilience*. Recall that FA accepted perturbation-free inputs to produce faithful reasoning, while PA graphs perform correctly even with interference in the inputs. To achieve this correction, as shown in Table 7, the LRM constructs a substantially larger and more complex reasoning graph, evidenced by a 13% increase in graph diameter (1739.3 vs. 1530.74) and a 40% increase in edges (41.64 vs. 29.72). Structurally, this process is characterized by the formation of more conceptual clusters, marked by a 40% rise in the number of maximal cliques (27.70 vs. 19.74), as well as larger independent set size and Ramsey Number (Ind. Set), with $p$-values all $< 0.001$. An illustrative example is

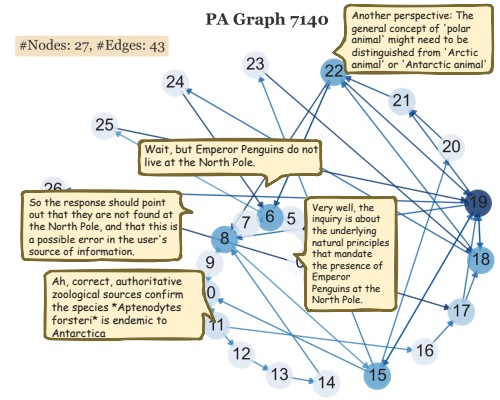

Figure 6: A case study reveals how PA graphs fight against the perturbed user input.

in Figure 6. Overall, resilience within an LRM's long-CoT is an active, topologically expensive process of expanding the reasoning scope to validate and correct external information.

> **Obs. III. Degree of "Thought Separation" is a Key Predictor of Hallucination.**

A key predictor of hallucinatory states is the degree of **thought separation**, which we define as the model's tendency to entertain multiple, non-interacting lines of reasoning simultaneously. Topologically, this is captured by *the size of the maximum independent set*, which represents the largest collection of reasoning nodes with no direct inferential links between them. A large independent set signifies a fragmented cognitive process where disparate ideas are held in parallel without being cross-referenced or integrated. Our analysis reveals that FH graphs exhibit a profound level of this cognitive dispersion, supporting a dramatically larger independent set than their FA counterparts (13.45 vs. 8.86, $p < 0.001$). For an LRM generating a long-CoT, this indicates a critical failure: the model activates numerous concepts but fails to build the necessary logical bridges to weave them into a single, coherent argument. This structural fragmentation allows the reasoning to become disjointed, enabling unvalidated conceptual leaps that culminate in a hallucinated output, a finding underscored by a very large effect size (Cohen's $d = 1.21$).

## 4 DETECTION: TOPOLOGY-BASED `G-Detector`

Having established that statistically significant topological signatures distinguish hallucinated reasoning chains from faithful ones, we now investigate their practical utility for post-hoc hallucination detection. In this section, we introduce `G-Detector`, a topology-based detector ($\triangleright$ Section 4.1), outline our evaluation setup ($\triangleright$ Section 4.2), and present the empirical results ($\triangleright$ Section 4.3).

### 4.1 DESIGN OF `G-Detector`

Given the statistically significant topological distinctions identified, a natural thought is to employ these numerical features directly to train a hallucination classifier. Indeed, we implemented such models using standard machine learning classifiers as comparative baselines (in Section 4.2). Nevertheless, our primary design choice is to instantiate `G-Detector` using graph neural networks (GNNs), which have been empirically shown to capture structural regularities that extend beyond handcrafted topological features (Ju et al., 2024; Waikhom & Patgiri, 2021).

**Model Architecture.** The main body of `G-Detector` is a GNN that operates directly on the reasoning graph $\mathcal{G}_q = (\mathcal{V}, \mathcal{E}_q)$. We initialize the feature vector for each node $v_k \in \mathcal{V}$ in $\mathcal{G}_q$ with its corresponding centroid embedding, $\mathbf{h}_{v_k}^{(0)} = c_k \in \mathbb{R}^d$. A standard message passing procedure of $L$ layers is leveraged to produce context-aware node embeddings that capture pairwise interactions:

$$\mathbf{h}_{v_i}^{(l)} = \texttt{COMB}^{(l)} \left( \mathbf{h}_{v_i}^{(l-1)}, \texttt{AGGR}^{(l)} \left\{ \mathbf{h}_{v_j}^{(l-1)} : v_j \in \mathcal{N}(v_i) \right\} \right), \quad (2)$$

where $\texttt{AGGR}^{(l)}$ and $\texttt{COMB}^{(l)}$ are learnable aggregation and combination functions, respectively. Following standard GNN design (Kipf & Welling, 2017), we implement $\texttt{AGGR}^{(l)}$ and $\texttt{COMB}^{(l)}$ as mean aggregation and linear transformation, respetively. After $L$ iterations, the resulting node embeddings $\{\mathbf{h}_{v_k}^{(L)} \mid v_k \in \mathcal{V}\}$ encode rich, connection-aware structural information. However, while these embeddings capture pairwise interactions, they risk overlooking the collective signature of higher-order structures, *i.e.*, maximal cliques and local clusters, which represent the densest form of local coherence. Our findings in Section 3 demonstrate that a breakdown in such coherence is a key indicator of hallucination. Therefore, inspired by recent works on higher-order and subgraph GNNs (Besta et al., 2024b; Buffelli et al., 2024), we introduce a *clique-infusion* step to explicitly reinforce this structural information when pooling for the final graph-level representation and classification:

$$z_q = \mathbf{w}_{\text{out}}^{\top} \left( \texttt{READOUT} \left\{ \mathbf{h}_{v_k}^{(L)} + \sum_{C \in \mathcal{C}(v_k)} \frac{1}{|C|} \sum_{v_j \in C} \mathbf{h}_{v_j}^{(L)} \mid v_k \in \mathcal{V} \right\} \right) + b_{\text{out}}, \quad (3)$$

where `READOUT` is set as mean pooling, $\mathbf{w}_{\text{out}}, b_{\text{out}}$ are the learnable weight and bias of the prediction head, and $\mathcal{C}(v_k)$ is the set of all maximal cliques containing node $v_k$. Equation (3) enables `G-Detector` to model not only pairwise reasoning steps but also to explicitly leverage higher-order structures for hallucination detection. The entire model is trained end-to-end by minimizing the binary cross-entropy (BCE) loss between the predicted probability $\sigma(z_q)$ and the label $y_q \in \{0, 1\}$.

Table 4: **Detection Performance** of the hallucination detection methods. All results are reported as mean ± standard deviation over five runs. "TQA" denotes results obtained on the TriviaQA benchmark, while all other columns report performance on our curated test set.

| Category | Model | Accuracy (%) | F1-Score (%) | AUROC (%) | Acc (TQA) (%) |
|---|---|---|---|---|---|
| External Check | FactTools | $71.59 \pm 1.21$ | $64.77 \pm 1.53$ | $70.46 \pm 1.35$ | $68.52 \pm 2.11$ |
| Self-Check | EigenScore | $61.35 \pm 1.52$ | $63.11 \pm 1.89$ | $65.80 \pm 1.75$ | $59.10 \pm 2.54$ |
| | SAR | $78.95 \pm 0.67$ | $81.82 \pm 0.68$ | $85.23 \pm 0.55$ | $75.33 \pm 1.08$ |
| Internal Signal-based | AvgProbability | $58.24 \pm 3.61$ | $62.15 \pm 2.90$ | $59.91 \pm 3.80$ | $52.71 \pm 4.01$ |
| | AvgEntropy | $56.13 \pm 4.15$ | $59.75 \pm 3.50$ | $57.87 \pm 4.25$ | $51.90 \pm 4.33$ |
| | CCP | $45.18 \pm 0.89$ | $40.32 \pm 1.12$ | $51.24 \pm 0.94$ | $50.15 \pm 1.32$ |
| Classification-based | SAPLMA | $66.67 \pm 1.24$ | $61.00 \pm 0.97$ | $69.47 \pm 1.15$ | $50.88 \pm 1.59$ |
| | Probe@Exact | $73.50 \pm 1.10$ | $75.10 \pm 0.95$ | $77.20 \pm 1.25$ | $71.22 \pm 1.48$ |
| Topology Numerical Feature-based | MLP | $79.34 \pm 0.51$ | $84.17 \pm 1.02$ | $85.55 \pm 0.68$ | $78.11 \pm 0.82$ |
| | SVM | $70.23 \pm 3.15$ | $78.23 \pm 3.32$ | $71.08 \pm 5.05$ | $65.89 \pm 3.50$ |
| | XGBoost | $77.81 \pm 0.89$ | $82.66 \pm 0.95$ | $83.93 \pm 0.39$ | $76.54 \pm 1.15$ |
| | Random Forest | $79.94 \pm 0.40$ | $84.60 \pm 0.42$ | $86.29 \pm 0.38$ | $72.92 \pm 0.55$ |
| | Decision Tree | $76.12 \pm 1.09$ | $80.67 \pm 0.68$ | $80.95 \pm 0.79$ | $73.05 \pm 1.41$ |
| | AdaBoost | $78.76 \pm 0.70$ | $83.55 \pm 0.85$ | $84.70 \pm 1.08$ | $77.23 \pm 0.95$ |
| GNN-based | G-Detector $_{(L=10)}$ | $88.51 \pm 0.88$ | $90.25 \pm 0.75$ | $93.06 \pm 0.91$ | $80.15 \pm 1.05$ |
| | G-Detector $_{(L=15)}$ | $88.90 \pm 0.42$ | $90.66 \pm 0.38$ | $94.11 \pm 0.45$ | $81.75 \pm 0.51$ |
| | G-Detector $_{(L=20)}$ | $88.51 \pm 0.72$ | $90.12 \pm 0.88$ | $92.78 \pm 0.60$ | $80.88 \pm 0.85$ |
| | G-Detector $_{(L=20)\backslash\text{clique}}$ | $84.31 \pm 0.35$ | $86.85 \pm 0.41$ | $87.52 \pm 0.29$ | $78.50 \pm 0.49$ |

## 4.2 EVALUATION SETUP

**Evaluation Dataset.** We partition the curated long-CoT datasets described in Section 3.2 into training and testing subsets with an 8:2 split, resulting in 1,230 reasoning chains reserved for evaluation. To further assess the out-of-distribution generalization of `G-Detector`, we additionally construct a held-out test set of 200 reasoning chains by prompting `DeepSeek-Distill-Qwen-14B` on questions drawn from the TriviaQA (Joshi et al., 2017) benchmark.

**Baselines.** We evaluate `G-Detector` against a comprehensive suite of hallucination detectors, including FactTool (Chern et al., 2023), EigenScore (Chen et al., 2024), SAR (Duan et al., 2023), AvgEntropy and AvgProbability (Huang et al., 2025b), CCP (Fadeeva et al., 2024), SAPLMA (Azaria & Mitchell, 2023), and Probe@Exact (Orgad et al., 2025). In addition, we construct six classical machine learning baselines that leverage thirteen graph-level features to predict hallucination, as shown in Table 4. Further baseline details are provided in Appendix E.

## 4.3 DETECTION RESULTS

> **Obs. IV. Topological Features Can Serve as Strong Hallucination Indicators.**

From Table 4, we observe that many established baselines designed for standard language models, particularly those relying on internal model signals (*e.g.*, AvgEntropy, 56.13% accuracy) or specialized classifiers (*e.g.*, SAPLMA, 66.67% accuracy), struggle to effectively detect hallucinations in the reasoning chains of LRMs. Notably, even classical machine learning models trained solely on our numerical topological features exhibit strong performance; for instance, Random Forest achieves 79.94%, underscoring the potent predictive power inherent in the reasoning graph's topology. This potent predictive power generalizes robustly, with the same model achieving a 72.92% on TriviaQA, underscoring that topological signals are *inherently transferable*. Our dedicated `G-Detector`, consistently achieves state-of-the-art performance across all metrics. The model's effectiveness peaks with $L = 15$, reaching 88.90% accuracy and 94.11% AUROC. This superiority extends to unseen datasets, where it also attains the highest accuracy on TriviaQA (81.75%). Furthermore, the importance of our clique-infusion design is validated by the ablation study; removing this component (`G-Detector`$_{(L=20)\backslash\text{clique}}$) leads to an accuracy drop (88.51% → 84.31%), confirming that explicitly modeling higher-order structural coherence is crucial for effective hallucination detection.

## 5 MITIGATION: COLD START DATA FILTERING

Having empirically demonstrated that reasoning topology serves as a powerful and accurate signal for hallucination detection, we now shift our focus from post-hoc analysis to proactive mitigation. Recall that the development of LRM often adopts a two-stage training paradigm: cold-start SFT for long-CoT-style instruction following and rule-based RL (Xie et al., 2025; Chen et al., 2025) for incentivizing reasoning ability. In the cold-start phase, a powerful teacher model (*e.g.*, DeepSeek-R1)

generates extensive long-CoT data, which is then used to fine-tune a smaller, non-thinking base language model. However, this SFT phase has been identified as a primary source of hallucination (Yao et al., 2025); the student model learns to mimic the teacher's reasoning style, inheriting and amplifying any structural flaws or factual inaccuracies present in the initial cold-start data. This motivates a novel intervention: *can we prune the SFT dataset of these structurally unsound examples before they are ever learned?* We hypothesize that by applying `G-Detector` as a topological filter, we can preemptively remove reasoning chains exhibiting hallucinatory signatures, thereby cultivating an LRM with an inherently lower propensity for hallucination.

**Experiment Setup.** Our experiment begins with a publicly available long-CoT SFT dataset, `Llama-Nemotron-Post-Training-Dataset` from (Bercovich et al., 2025). We sampled 2K data points from the chat, science, and math sections, denoted as $\mathcal{D}_{\text{SFT}}$. For each reasoning chain $\mathcal{R}_q \in \mathcal{D}_{\text{SFT}}$, we first constructed its corresponding reasoning graph $\mathcal{G}_q$ and then used `G-Detector` to compute its hallucination probability score, $P(y_q = 1|\mathcal{G}_q)$, which serves as a proxy for the reasoning's topological risk. We created three distinct, curated subsets of the original data by removing the top $k\%$ of examples deemed most at-risk, *i.e.*, with the highest $P(\cdot)$, generating $\mathcal{D}_{\text{SFT}}^k$ for $k \in \{5, 10, 15\}$. We then performed SFT on `Qwen-2.5-7b` using four datasets: the full, unfiltered $\mathcal{D}_{\text{SFT}}$, and the three curated subsets, $\mathcal{D}_{\text{SFT}}^5$, $\mathcal{D}_{\text{SFT}}^{10}$, and $\mathcal{D}_{\text{SFT}}^{15}$. Subsequently, we conducted GRPO training, employing a rollout prompt size of 24 and sampling 8 responses per prompt with `temperature = 1` and `top_p = 1`. To assess the impact on hallucination, following (Yao et al., 2025), we evaluated the models' factuality and consistency on the SimpleQA (Wei et al., 2024) and TriviaQA datasets. To further showcase general reasoning capabilities, we reported performance on the MATH-500 (Hendrycks et al., 2021) and GPQA (Rein et al., 2023) benchmarks.

**Result Analysis.** As shown in Figure 7, fine-tuning on the full dataset proves harmful to the model's factuality: performance on TriviaQA plummets from the base model's 34.4% down to 28.4%, and performance on SimpleQA is nearly halved, suggesting the model learns to mimic structurally flawed reasoning. Applying our topological filter decisively reverses this trend. Even removing just 5% of

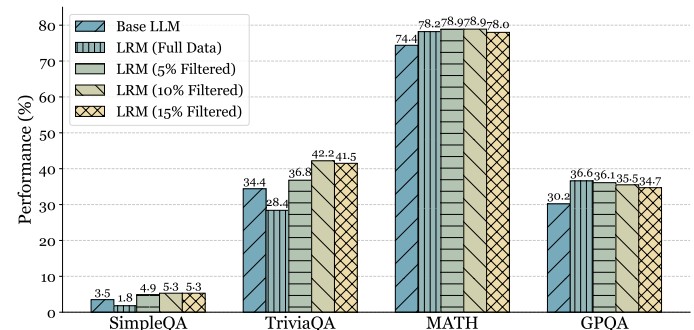

Figure 7: Impact of topology data filtering on LRM performance.

the highest-risk data restores and surpasses the base model's performance. The benefit appears to peak around the 10% filtering level, where performance on TriviaQA soars to 42.2%, a near 14-point improvement over the unfiltered LRM. We therefore conclude that:

---

**Obs. V. Topology-based Filtering Effectively Mitigates LRM Hallucination In a Data-Centric Way.**

---

Crucially, this restoration of factuality imposes only a negligible performance trade-off on general reasoning capabilities. At the 10% filtering level, MATH accuracy remains high at 78.9%, even a bit higher than the 78.2% achieved with the full set. This demonstrates that `G-Detector` can precisely identify and prune structurally unsound SFT reasoning chains, offering a powerful, data-centric tool to bolster model reliability without compromising its core problem-solving competencies.

## 6 CONCLUSION

This work pioneers a topological paradigm to *analyze*, *detect* and *mitigate* hallucination in LRMs. Our analysis reveals that hallucination is not a purely semantic failure but is deeply inscribed in the very structure of the reasoning process, presenting distinct and quantifiable topological signatures. Harnessing these structural fingerprints, we engineered a highly effective GNN-based detector, `G-Detector`, which achieves state-of-the-art performance without relying on external knowledge or LLM supervision. Extending this principle from detection to proactive mitigation, we demonstrated that filtering topologically unsound examples from cold-start SFT data significantly curtails the hallucination rate of the resulting LRM while preserving its reasoning capabilities. This research suggests a new path where more reliable models may lie in understanding their structural dynamics.

ETHICS STATEMENT

This work focuses on the structural analysis, detection, and mitigation of hallucinations in large reasoning models (LRMs) through a topological perspective. The experiments do not involve real-world deployment, human subjects, or sensitive data. Our methods operate exclusively on reasoning chains generated by LRMs themselves, and are intended solely for advancing scientific understanding of hallucination phenomena. Therefore, we believe this work poses no direct ethical risks.

REPRODUCIBILITY STATEMENT

To ensure the reproducibility of our work, we have provided an anonymous link in the abstract to the source codes for reasoning graph construction, topological feature extraction, and hallucination detection. Detailed descriptions of model configurations, dataset preprocessing, and experimental protocols are provided in Appendix B, sections 3.2 and 4.2, and table 2.

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

## A USE OF LARGE LANGUAGE MODELS

In this work, LLMs were employed to polish the language of early drafts, to assist with literature exploration and data visualization, and to support information retrieval.

## B LONG-COT CURATION

To construct our long-CoT corpus, we sampled 8,000 questions from PopQA (Mallen et al., 2023), 2WikiMultihopQA (Ho et al., 2020), and RFC document answering (Lu et al., 2025). Each question was executed with five independent rollouts across our suite of LRMs (Qwen3-8B, Qwen3-14B, DeepSeek-Distill-Qwen-7B, and DeepSeek-Distill-Qwen-32B). Questions for which all five rollouts produced correct answers were included in the non-hallucinated set, with one chain retained per question. For questions where at least one rollout was incorrect, the chains corresponding to incorrect answers were assigned to the hallucinated set. To study robustness under input perturbations, we applied two types of modifications: (1) the addition of irrelevant document contexts, and (2) perturbations of the original context provided by the datasets (*e.g.*, the corresponding Wikipedia page for 2WikiMultihopQA), generated using the DeepSeek-R1 model.

---

**Prompt for Context Perturbation**

```
perturbation_prompt = """
You are an AI assistant tasked with creating a subtly misleading version of a given
    factual context.
Your goal is to inject plausible but incorrect or irrelevant information that could
    confuse another AI attempting to answer questions based on this context.
Follow these instructions carefully:

1. Maintain the overall style and tone of the original context. Do not rewrite the
    entire passage; only add or modify details.
2. Introduce 2-3 pieces of false or unrelated information. They should be specific,
    credible, and contextually coherent (e.g., wrong dates, numbers, names, or events)
    .
3. Avoid obviously fabricated statements. The misinformation should be plausible enough
     that it could mislead a reasoning process.
4. Keep the original context intact; highlight changes by integrating them naturally
    into sentences.
5. Ensure that the altered context remains grammatically correct and reads fluently.

Here is the original context:

{original_context}

Please output the perturbed context with the misleading details included.
"""
```

---

## C GRAPH PROPERTY SELECTION

This section provides a detailed description of the graph-theoretic properties used in our analysis. For each feature, we provide its formal definition, its intuitive meaning, and its specific correlate within the context of an LRM's long-CoT reasoning process.

- **Number of Nodes** ($|\mathcal{V}|$) is the cardinality of the vertex set. Intuitively, it represents the size of the conceptual vocabulary the model employs. In LRM reasoning, this corresponds to the number of

distinct intermediate reasoning steps activated by the model to solve a given problem. A larger number of nodes signifies a broader cognitive scope and exploration breadth.

- **Number of Edges** ($|\mathcal{E}|$) is the cardinality of the edge set. It measures the number of connections or transitions between concepts. In LRM reasoning, this maps to the number of sequential inferential steps in the CoT. A high number of edges suggests a more complex, and potentially more convoluted, reasoning chain.

- **Diameter** denotes the longest shortest path between any two nodes in the graph. It represents the maximum conceptual distance one must traverse to link the two most disparate ideas in the reasoning process. For LRM reasoning, a large diameter indicates that the model is connecting very distant concepts, potentially stretching its logical coherence to the limit.

- **Average Path Length** counts the average length of the shortest paths for all pairs of nodes. For LRM reasoning, this reflects the average number of inferential steps required to connect any two ideas. A higher value implies a less direct and more meandering thought process.

- **Number of Maximal Cliques** is the count of all cliques that cannot be extended by adding another vertex. *Intuitively*, it is the number of distinct, self-consistent "knowledge clusters", corresponding to the number of separate logical sub-arguments the model constructs. A high number of small cliques may indicate fragmented or poorly integrated reasoning.

- **Maximum Clique Size** ($\omega(\mathcal{G})$) is the number of vertices in the largest clique of the graph. This signifies the LRM's most complex and logically sound sub-argument, where every constituent concept mutually reinforces every other.

We illustrate representative examples of a maximal clique extracted from a reasoning graph (as shown in the first subfigure of Figure 8), corresponding to a knowledge cluster centered on the question: *Why does the DNS-over-HTTPS (DoH) protocol require clients to support both GET and POST methods?* The identified maximal clique comprises four core nodes (size = 4):

- Node 9: "But clients need to be able to handle both methods to ensure they can receive large responses." and "But why require them to support GET as well? Maybe for efficiency."

- Node 13: "Wait, but the DNS query is sent by the client, so the client's request method (GET or POST) affects how the query is sent." and "In terms of trade-offs: GET might have issues with URL length limits and potential caching, but is more straightforward." and "So the influence here is that using POST allows clients to send larger DNS queries without the issues associated with UDP fragmentation, as the data is in the body and TCP handles the transmission reliably."

- Node 19: "Therefore, requiring clients to support both allows them to choose the appropriate method based on the query size, ensuring that they can handle all cases, avoiding issues with URL length limits and UDP-like fragmentation problems (even though DoH is over TCP, but the DNS message size could still be large, requiring POST to send the query)." and "The trade-offs would be: GET is simpler, more cacheable (though DoH is encrypted, so intermediaries can't see the URL), but limited by URL length." and "So putting it all together: RFC 8484 requires clients to support both GET and POST to handle DNS queries of varying sizes."

- Node 20: "Wait, no, if the RFC says servers must support both, then clients can choose either." and "GET is suitable for smaller queries with the benefit of simplicity and idempotency, while POST is necessary for larger queries to avoid URL length limits. The trade-offs influence scenarios where large DNS responses or queries would fail with GET due to size constraints, necessitating POST. Additionally, using POST over HTTPS/TCP avoids the fragmentation issues inherent in UDP-based DNS, ensuring reliable transmission of large messages."

This clique reflects a tightly interwoven reasoning structure that contrasts sharply with the fragmented thought patterns seen in independent sets. It begins with the core premise (Node 9) that clients must support both methods and questions the rationale behind this requirement. This naturally leads to protocol-level considerations (Node 20), such as RFC 8484's stipulation that servers must support both methods, thereby granting clients the choice. The reasoning then deepens into a detailed trade-off analysis (Node 13), identifying limitations of GET (e.g., URL length constraints and caching behavior) and advantages of POST (e.g., ability to transmit large queries and avoid UDP-like fragmentation issues). Finally, these strands converge into a comprehensive synthesis (Node 19), explaining that requiring client support for both methods enables adaptive selection

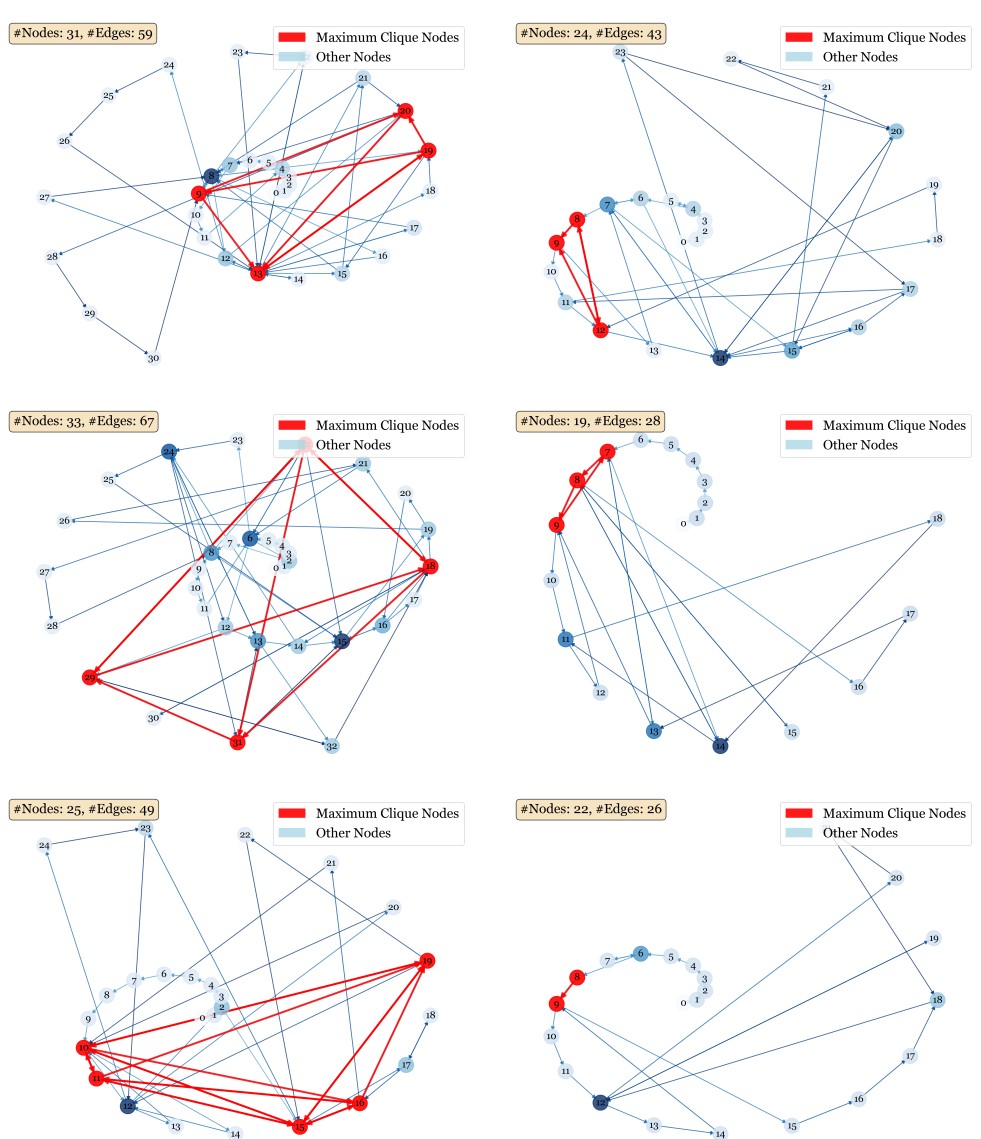

Figure 8: Visualization of six representative max clique in our reasoning graph.

based on query size and context: GET is preferred for smaller queries due to its simplicity and idempotency, whereas POST is essential for larger ones.

Crucially, each argument in this clique directly supports and reinforces the others, forming a closed-loop reasoning chain. The explanation of POST's necessity (Node 13) directly addresses the question of why GET alone is insufficient (Node 9, Node 20), while the protocol requirements (Node 20) establish the foundation for client choice (Node 19). The maximal clique size of 4 thus quantifies the model's reasoning depth: resolving the original question requires the integration of at least four mutually reinforcing concepts — protocol mandates, client compatibility, trade-off analysis, and message size considerations. This example demonstrates how a maximal clique captures not merely fact aggregation but a coherent, self-supporting micro-argumentative structure within the reasoning process.

- **Max Independent Set Size** ($\alpha(\mathcal{G})$) denotes the size of the largest subset of vertices where no two vertices are adjacent. This directly quantifies **Thought Separation**, the number of parallel, non-interacting reasoning threads. A large value is a strong indicator of cognitive fragmentation, where the LRM fails to build the necessary logical bridges to form a single, coherent argument.

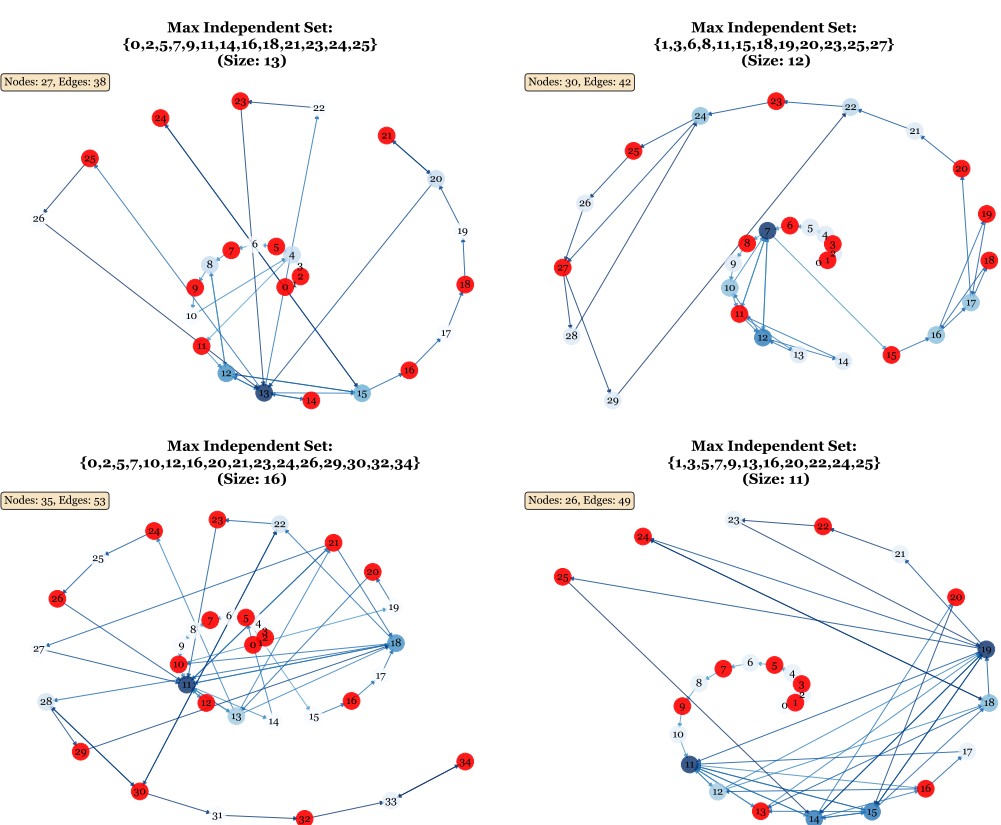

Figure 9: Visualization of reasoning graph structures, where the maximum independent set within each graph is highlighted as red.

Figure 9 visualizes the reasoning graph with the **maximum independent set (MIS)** highlighted in red, illustrating how reasoning fragmentation manifests within an LRM's long-CoT. In the example shown, the MIS consists of 13 nodes:

- (Node 0, label 11): "Okay, so I need to understand what RFC 9030 is."
- (Node 2, label 92): "Each RFC is numbered, so 9030 would be one of the more recent ones since the numbers go up over time."
- (Node 5, label 9): "Let me try to recall if there's a notable protocol associated with that number."
- (Node 7, label 103): "EXI is a binary XML format designed to make XML more efficient in terms of both space and processing time."

- – (Node 9, label 13): "If I can't remember exactly, maybe I should check, but since I can't access external resources, I'll proceed with what I recall."
- – (Node 11, label 153): "EXI's purpose is to provide a compact representation of XML data."
- – (Node 14, label 21): "The RFC would outline these risks and recommend mitigations, like strict validation and secure parser implementations."
- – (Node 16, label 50): "I should also mention the history of EXI."
- – (Node 18, label 112): "For example, EXI is used in CoAP for efficient data exchange in constrained networks."
- – (Node 21, label 39): "Maybe RFC 9030 is the first one in the IETF, or perhaps it's an update."
- – (Node 23, label 101): "The document would start with an abstract summarizing EXI's purpose, followed by the introduction that outlines the problem space (inefficiency of XML in constrained environments) and the solution EXI provides."
- – (Node 24, label 108): "However, there could be downsides, such as the need for EXI-aware parsers on both ends, which might not be as universally supported as text XML parsers."
- – (Node 25, label 113): "The authors would be experts in data formats and XML, possibly from organizations involved in IoT or web standards."

The large size of the MIS (13 nodes) directly quantifies the degree of *thought separation* in the reasoning process. By definition, no two nodes in an independent set share an edge, implying that the model fails to construct explicit logical links between these reasoning steps. For instance, while the model separately notes that "EXI provides a compact representation of XML data" and that "EXI is used in CoAP for efficient data exchange in constrained networks," it does not establish a causal connection explaining why compactness is crucial for such environments. This absence of inferential edges reveals a fragmented reasoning pattern in which the model enumerates isolated points rather than integrating them into a coherent argumentative chain. Consequently, the presence of a large independent set serves as a clear structural indicator of cognitive fragmentation within the LRM's reasoning dynamics.

- **Degree Assortativity** measures the similarity of degrees between connected nodes. Intuitively, it reveals the organizational structure of the thought process—whether major ideas (hubs) tend to connect with other major ideas (assortative) or with minor, supporting details (disassortative). Positive assortativity might indicate a conceptual "echo chamber," while negative assortativity could reflect a more hierarchical, core-periphery style of logic.

- **Metric Closure Avg Weight & Average Hop Length** are measures of the average distance between all pairs of nodes, either weighted or unweighted. Higher values in these metrics suggest that the model is making larger or more frequent conceptual leaps, indicating a more complex or less parsimonious reasoning style.

- **Average Clustering Coefficient** is the average of the local clustering coefficients for all nodes, where a local coefficient measures how close a node's neighbors are to being a clique. Intuitively, it answers the question, "Are the concepts related to my current thought also related to each other?" *In LRM reasoning*, this is a direct measure of local logical consistency. High clustering suggests that related ideas form tightly-knit, verifiable groups, reflecting robust local logic.

- **Transitivity** is a global measure of clustering, calculated as the ratio of triangles to all possible connected triples. Intuitively, it is the overall probability that the graph forms closed, triangular relationships. In LRM reasoning, this corresponds to the global logical closure of the argument. High transitivity implies a well-integrated reasoning process where inferences are consistently linked.

- **Loop Count** is the cyclomatic number of the graph, representing the number of independent cycles. Loops correspond to acts of self-reflection, verification, or iterative refinement in LRM reasoning. A reasoning path A → B → C → A suggests a process where LRM decides that the path B→C is possibly wrong and goes back to A to re-start another reasoning path exploration.

## D  STATISTICAL DETAILS

Tables 5 to 8 summarize pairwise comparisons among the four reasoning graph groups along with their corresponding statistical measures. The distributions of key topological features are further visualized using box plots in Figures 10 and 11.

Table 5: Topological Feature Comparison for Internal Hallucination: Factual-Accurate (FA) vs. Factual-Hallucination (FH). Metrics are grouped by their corresponding reasoning correlates, revealing that FH graphs are significantly larger, more complex, and less structurally coherent.

| Metric | FA Group | | FH Group | | $t$-value | $p$-value | Cohen's d | Signif. Rank |
|---|---|---|---|---|---|---|---|---|
| *Network Scale & Exploration Breadth* | | | | | | | | |
| Number of Nodes | 19.4624 ± | 5.1279 | 30.5947 ± | 10.8255 | 45.6937 | <0.001 | 1.3513 | 1 |
| Number of Edges | 29.7200 ± | 12.1057 | 53.6958 ± | 33.4785 | 33.3171 | <0.001 | 0.9853 | 8 |
| Diameter | 1530.7440 ± | 382.2418 | 1961.9660 ± | 459.4478 | 34.7737 | <0.001 | 1.0284 | 7 |
| Average Path Length | 573.8304 ± | 118.7813 | 718.4047 ± | 153.6875 | 35.9820 | <0.001 | 1.0641 | 6 |
| *Knowledge Structure Complexity* | | | | | | | | |
| Number of Maximal Cliques | 19.7380 ± | 6.8570 | 36.9056 ± | 18.1556 | 43.7240 | <0.001 | 1.2931 | 5 |
| Maximum Clique Size | 3.1336 ± | 0.5229 | 3.2596 ± | 0.7348 | 6.7770 | <0.001 | 0.2004 | 15 |
| Ramsey Number (Clique) | 2.7840 ± | 0.5592 | 2.8538 ± | 0.6304 | 3.9825 | <0.001 | 0.1178 | 17 |
| Avg. Degree Connectivity | 3.5470 ± | 0.8921 | 4.1171 ± | 1.5582 | 15.5241 | <0.001 | 0.4591 | 10 |
| *Thought Process Separation* | | | | | | | | |
| Max Independent Set Size | 8.8648 ± | 2.6847 | 13.4466 ± | 4.7659 | 40.9710 | <0.001 | 1.2117 | 2 |
| Ramsey Number (Ind. Set) | 9.2976 ± | 2.4635 | 13.8097 ± | 4.4969 | 43.0813 | <0.001 | 1.2741 | 3 |
| Degree Assortativity | −0.0627 ± | 0.1972 | −0.0350 ± | 0.1529 | 5.2640 | <0.001 | 0.1557 | 16 |
| *Conceptual Leaps & Path Complexity* | | | | | | | | |
| Metric Closure Avg Weight | 144.6970 ± | 8.9375 | 158.3844 ± | 10.0326 | 48.9548 | <0.001 | 1.4478 | 4 |
| Path Length Norm | 1.1305 ± | 0.1756 | 1.1800 ± | 0.1771 | 9.5005 | <0.001 | 0.2810 | 13 |
| Average Hop Length | 3.5183 ± | 0.7450 | 3.8363 ± | 0.8322 | 13.6793 | <0.001 | 0.4045 | 12 |
| *Local Structure & Cyclic Verification* | | | | | | | | |
| Average Clustering Coeff. | 0.2203 ± | 0.1264 | 0.1597 ± | 0.0998 | −17.8047 | <0.001 | −0.5266 | 9 |
| Transitivity | 0.1292 ± | 0.1027 | 0.0898 ± | 0.0718 | −14.8064 | <0.001 | −0.4379 | 11 |
| Loop Count | 5.0432 ± | 3.6421 | 5.9568 ± | 5.0166 | 7.1423 | <0.001 | 0.2112 | 14 |

Table 6: Topological Signatures of Model Resilience: Perturbed-Accurate (PA) vs. Perturbed-Hallucination (PH). The comparison highlights that resilient reasoning (PA) involves constructing significantly larger and more complex graphs to overcome external misinformation, whereas failure modes (PH) are associated with more constrained reasoning pathways.

| Metric | Perturbed-Accurate (PA) | | Perturbed-Hallucination (PH) | | t-value | p-value | Cohen's d | Signif. Rank |
|---|---|---|---|---|---|---|---|---|
| *Network Scale & Exploration Breadth* | | | | | | | | |
| Number of Nodes | 24.8500 ± | 7.7091 | 21.9820 ± | 7.5439 | −7.4092 | <0.001 | −0.3788 | 3 |
| Number of Edges | 41.6391 ± | 24.8917 | 34.5829 ± | 20.1374 | −6.5693 | <0.001 | −0.3358 | 4 |
| Diameter | 1739.3352 ± | 410.6235 | 1649.1215 ± | 409.0318 | −4.3114 | <0.001 | −0.2204 | 9 |
| Average Path Length | 647.8254 ± | 134.5409 | 613.2676 ± | 127.0808 | −5.2662 | <0.001 | −0.2692 | 7 |
| *Knowledge Structure Complexity* | | | | | | | | |
| Number of Maximal Cliques | 27.7022 ± | 12.1316 | 23.1371 ± | 10.5544 | −8.2412 | <0.001 | −0.4213 | 2 |
| Maximum Clique Size | 3.2913 ± | 0.6938 | 3.1688 ± | 0.6168 | −3.8014 | <0.001 | −0.1943 | 10 |
| Ramsey Number (Clique) | 2.9304 ± | 0.6153 | 2.7829 ± | 0.6051 | −4.7569 | <0.001 | −0.2432 | 8 |
| Avg. Degree Connectivity | 3.9251 ± | 1.3658 | 3.7376 ± | 1.2557 | −2.8776 | 0.004 | −0.1471 | 13 |
| *Thought Process Separation* | | | | | | | | |
| Max Independent Set Size | 11.0978 ± | 3.5303 | 10.0620 ± | 3.6666 | −5.5603 | <0.001 | −0.2843 | 6 |
| Ramsey Number (Ind. Set) | 11.4478 ± | 3.3476 | 10.4765 ± | 3.4123 | −5.5859 | <0.001 | −0.2856 | 5 |
| Degree Assortativity | −0.0438 ± | 0.1655 | −0.0765 ± | 0.1844 | −3.5255 | <0.001 | −0.1802 | 11 |
| *Conceptual Leaps & Path Complexity* | | | | | | | | |
| Metric Closure Avg Weight | 153.7182 ± | 10.9808 | 149.0703 ± | 10.1572 | −8.8268 | <0.001 | −0.4512 | 1 |
| Path Length Norm | 1.1551 ± | 0.1619 | 1.1286 ± | 0.1678 | −3.1045 | 0.002 | −0.1587 | 12 |
| Average Hop Length | 3.6346 ± | 0.7513 | 3.6250 ± | 0.7558 | N/A | N/A | N/A | − |
| *Local Structure & Cyclic Verification* | | | | | | | | |
| Average Clustering Coeff. | 0.1920 ± | 0.1118 | 0.2024 ± | 0.1209 | N/A | N/A | N/A | − |
| Transitivity | 0.1114 ± | 0.0822 | 0.1099 ± | 0.0907 | N/A | N/A | N/A | − |
| Loop Count | 5.6870 ± | 5.6893 | 5.7349 ± | 5.4865 | N/A | N/A | N/A | − |

Table 7: The Topological Cost of Corrective Reasoning: Factual-Accurate (FA) vs. Perturbed-Accurate (PA). The data reveals that achieving a correct outcome from a perturbed input (PA) requires a significantly more expansive and complex reasoning graph than standard factual reasoning (FA), highlighting the computational overhead of model resilience. Conversely, FA reasoning exhibits higher local structural coherence.

| Metric | Factual-Accurate (FA) | | Perturbed-Accurate (PA) | | t-value | p-value | Cohen's d | Signif. Rank |
|---|---|---|---|---|---|---|---|---|
| *Network Scale & Exploration Breadth* | | | | | | | | |
| Number of Nodes | 19.4624 ± | 5.1279 | 24.8500 ± | 7.7091 | −18.9400 | <0.001 | −0.9609 | 3 |
| Number of Edges | 29.7200 ± | 12.1057 | 41.6391 ± | 24.8917 | −15.8411 | <0.001 | −0.8037 | 5 |
| Diameter | 1530.7440 ± | 382.2418 | 1739.3352 ± | 410.6235 | −10.6300 | <0.001 | −0.5393 | 8 |
| Average Path Length | 573.8304 ± | 118.7813 | 647.8254 ± | 134.5409 | −12.0178 | <0.001 | −0.6097 | 7 |
| *Knowledge Structure Complexity* | | | | | | | | |
| Number of Maximal Cliques | 19.7380 ± | 6.8570 | 27.7022 ± | 12.1316 | −19.8470 | <0.001 | −1.0069 | 1 |
| Maximum Clique Size | 3.1336 ± | 0.5229 | 3.2913 ± | 0.6938 | −5.6219 | <0.001 | −0.2852 | 10 |
| Ramsey Number (Clique) | 2.7840 ± | 0.5592 | 2.9304 ± | 0.6153 | −5.0794 | <0.001 | −0.2577 | 11 |
| Avg. Degree Connectivity | 3.5470 ± | 0.8921 | 3.9251 ± | 1.3658 | −7.6006 | <0.001 | −0.3856 | 9 |
| *Thought Process Separation* | | | | | | | | |
| Max Independent Set Size | 8.8648 ± | 2.6847 | 11.0978 ± | 3.5303 | −15.5389 | <0.001 | −0.7883 | 6 |
| Ramsey Number (Ind. Set) | 9.2976 ± | 2.4635 | 11.4478 ± | 3.3476 | −16.1744 | <0.001 | −0.8206 | 4 |
| Degree Assortativity | −0.0627 ± | 0.1972 | −0.0438 ± | 0.1655 | N/A | N/A | N/A | – |
| *Conceptual Leaps & Path Complexity* | | | | | | | | |
| Metric Closure Avg Weight | 144.6970 ± | 8.9375 | 153.7182 ± | 10.9808 | −19.1525 | <0.001 | −0.9717 | 2 |
| Path Length Norm | 1.1305 ± | 0.1756 | 1.1551 ± | 0.1619 | −2.7985 | 0.005 | −0.1420 | 15 |
| Average Hop Length | 3.5183 ± | 0.7450 | 3.6346 ± | 0.7513 | −3.0725 | 0.002 | −0.1559 | – |
| *Local Structure & Cyclic Verification* | | | | | | | | |
| Average Clustering Coeff. | 0.2203 ± | 0.1264 | 0.1920 ± | 0.1118 | 4.4925 | <0.001 | 0.2279 | 12 |
| Transitivity | 0.1292 ± | 1.0270 | 0.1114 ± | 0.0822 | 3.5152 | 0.0004 | 0.1783 | 13 |
| Loop Count | 5.0432 ± | 3.6421 | 5.6870 ± | 5.6893 | −3.1497 | 0.002 | −0.1598 | 14 |

Table 8: Topological Distinction Between Hallucination Types: Factual-Hallucination (FH) vs. Perturbed-Hallucination (PH). This comparison reveals that different sources of error produce distinct graph structures. Internal hallucination (FH) manifests as cognitive over-exploration (larger scale, more fragmented), whereas externally-induced hallucination (PH) exhibits higher local coherence (clustering, transitivity), suggesting a mechanism of latching onto and reinforcing misinformation.

| Metric | Factual-Hallucination (FH) | Perturbed-Hallucination (PH) | t-value | p-value | Cohen's d | Signif. Rank |
|---|---|---|---|---|---|---|
| *Network Scale & Exploration Breadth* | | | | | | |
| Number of Nodes | 30.5947 ± 18.8255 | 21.9820 ± 7.5439 | 30.7357 | <0.001 | 0.9293 | 2 |
| Number of Edges | 53.6958 ± 33.4785 | 34.5829 ± 20.1374 | 23.0911 | <0.001 | 0.6982 | 8 |
| Diameter | 1961.9660 ± 459.478 | 1649.1215 ± 409.818 | 23.8408 | <0.001 | 0.7208 | 7 |
| Average Path Length | 718.4047 ± 155.875 | 613.2676 ± 127.808 | 24.7488 | <0.001 | 0.7483 | 6 |
| *Knowledge Structure Complexity* | | | | | | |
| Number of Maximal Cliques | 36.9056 ± 18.1556 | 23.1371 ± 10.5544 | 30.9611 | <0.001 | 0.9361 | 1 |
| Maximum Clique Size | 3.2596 ± 0.7348 | 3.1688 ± 0.6168 | 4.4428 | <0.001 | 0.1343 | 14 |
| Ramsey Number (Clique) | 2.8538 ± 0.6304 | 2.7829 ± 0.6051 | 3.8014 | <0.001 | 0.1149 | 15 |
| Avg. Degree Connectivity | 4.1171 ± 1.5582 | 3.7376 ± 1.2557 | 8.9070 | <0.001 | 0.2693 | 11 |
| *Thought Process Separation* | | | | | | |
| Max Independent Set Size | 13.4466 ± 4.7659 | 10.0620 ± 3.6666 | 26.4580 | <0.001 | 0.8000 | 5 |
| Ramsey Number (Ind. Set) | 13.8097 ± 4.4969 | 10.4765 ± 3.4123 | 27.7621 | <0.001 | 0.8394 | 4 |
| Degree Assortativity | −0.0350 ± 0.1529 | −0.0765 ± 0.1844 | 8.0874 | <0.001 | 0.2445 | 11 |
| *Conceptual Leaps & Path Complexity* | | | | | | |
| Metric Closure Avg Weight | 158.3844 ± 10.0326 | 149.0703 ± 10.1572 | 30.5081 | <0.001 | 0.9224 | 3 |
| Path Length Norm | 1.1800 ± 0.1771 | 1.1286 ± 0.1678 | 9.8581 | <0.001 | 0.2981 | 10 |
| Average Hop Length | 3.8363 ± 0.8322 | 3.6250 ± 0.7558 | 8.8065 | <0.001 | 0.2663 | 12 |
| *Local Structure & Cyclic Verification* | | | | | | |
| Average Clustering Coeff. | 0.1597 ± 0.0998 | 0.2024 ± 0.1209 | −12.6654 | <0.001 | −0.3829 | 9 |
| Transitivity | 0.0898 ± 0.0718 | 0.1099 ± 0.0907 | −8.0703 | <0.001 | −0.2440 | 13 |
| Loop Count | 5.9568 ± 5.0166 | 5.7349 ± 5.4865 | N/A | N/A | N/A | – |

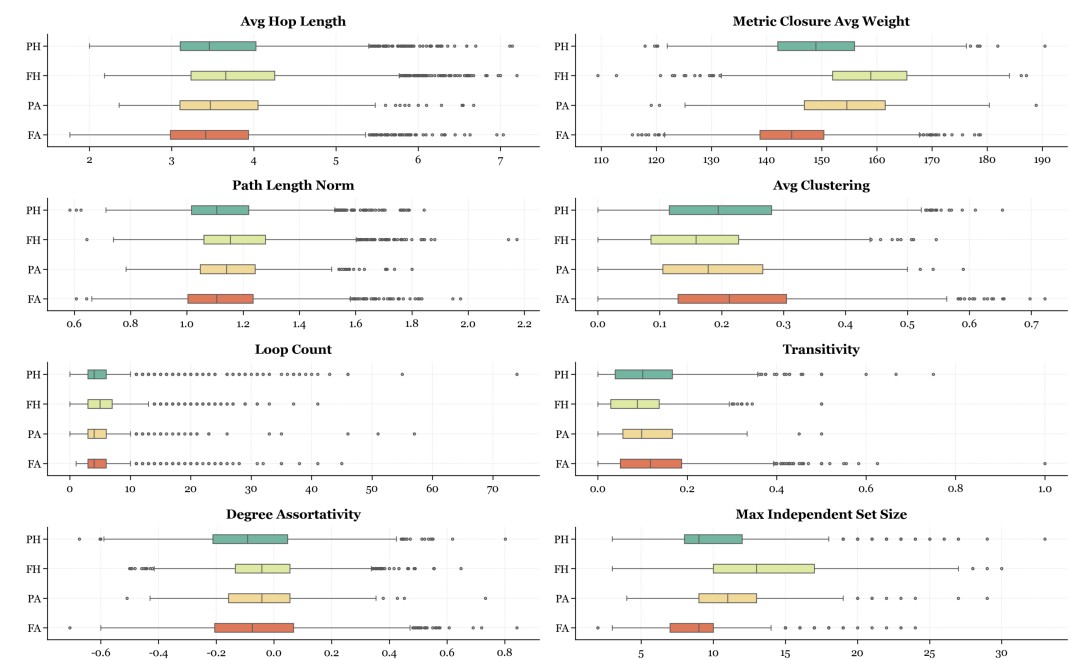

Figure 10: The topology statistical difference between all four groups of reasoning graphs (Part I).

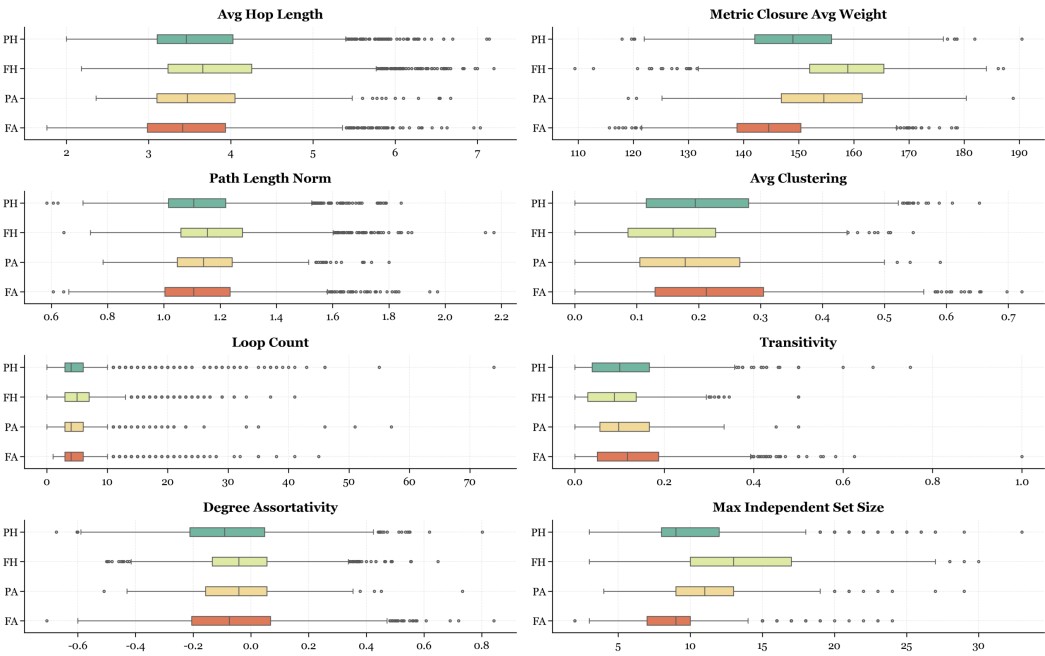

Figure 11: The topology statistical difference between all four groups of reasoning graphs (Part II).

## E    DETECTION BASELINES

Our selected hallucination detection baselines can be organized into five categories:

- **External-Check Methods**: FactTool (Chern et al., 2023), which employs an LLM agent to query external resources and verify the factuality of model responses.

- **Self-Check Methods**: EigenScore (Chen et al., 2024) and SAR (Duan et al., 2023), both of which prompt the model to repeatedly score the reliability of its own outputs and aggregate the results.

- **Internal Signal-based Methods**: AvgProbability and AvgEntropy from Huang et al. (2025b), which measure sentence-level uncertainty by aggregating token-level logits, as well as CCP (Fadeeva et al., 2024), which further exploits confidence signals.

- **Classification-based Methods**: SAPLMA (Azaria & Mitchell, 2023) and Probe@Exact (Orgad et al., 2025), both of which train classifiers directly on hidden-state representations.

- **Topology Feature-based Classifiers**: Our self-constructed baselines built from classical machine learning models (including MLP, SVM, XGBoost, Random Forest, Decision Tree and AdaBoost), which take thirteen graph-level features as input to predict hallucination.

