# OpenReview forum: "Unraveling Hallucination in Large Reasoning Models: A Topological Perspective"
_ICLR.cc/2026/Conference — Submitted to ICLR 2026_

### Official Review · Reviewer_5tYU · 2025-10-30

**Soundness:** 2
**Presentation:** 2
**Contribution:** 2
**Rating:** 2
**Confidence:** 5

**Summary:**

This paper proposes a topological framework for analyzing, detecting, and mitigating hallucinations in LRMs. The authors model each reasoning chain as a structured graph, where nodes represent reasoning states derived from hidden representations and edges capture transitions between reasoning steps, and statistically identify 17 graph-theoretic features that distinguish hallucinated from faithful reasoning. Based on these insights, the paper introduces G-Detector, a graph neural network-based post-hoc hallucination detector that leverages reasoning topology alone and reportedly achieves up to 88.9% accuracy and 94.1% AUROC. Finally, the authors extend G-Detector for data-centric mitigation by filtering high-risk reasoning traces during SFT, which improves factual accuracy on benchmarks such as SimpleQA and TriviaQA by around 13-14% without degrading reasoning ability. The work aims to reveal that hallucinations in LRMs are not random errors but exhibit identifiable structural signatures within reasoning topologies.

**Strengths:**

1. The paper introduces a new topological perspective on hallucination in LRMs. Unlike most prior work that treats hallucination as a token-level or factuality issue, this work models reasoning trajectories as structured graphs and studies their statistical and structural signatures.
2. The paper presents a coherent three-stage pipeline, (i) analyzing reasoning topology, (ii) detecting hallucination via G-Detector, and (iii) mitigating hallucination through topology-based filtering during SFT. This end-to-end narrative makes the study more complete than typical detection-only works.

**Weaknesses:**

1. The definition of symbol adopted in this paper is insufficiently clear; please refer to the detailed questions part below for specific concerns.
2. The paper lacks concrete examples. For instance, in the perturbation setting, it is unclear how the perturbed texts differ from the original inputs. According to the prompt shown in Appendix B, the perturbation process could easily cause the model to add irrelevant or tangential information, rather than producing contradictory or factually incorrect content. Furthermore, in the final SFT filtering stage, it would be highly informative to present several examples of the filtered samples, that is, which specific reasoning traces were considered "high risk" by G-Detector. Without such examples, the method remains largely a black box.
3. The paper does not clearly indicate on which models and datasets the reported findings are based. It is unclear whether the results are aggregated across multiple models and datasets or derived from a single specific configuration.
4. The definition of graph nodes in G-Detector is ambiguous. Do the nodes corresponding to the first reasoning step in different problems share the same representation? If so, this design is problematic, since "the first step" may carry entirely different semantic meanings across questions. If not, it implies that the model learns separate node embeddings for every question, which makes strong generalization unlikely.
5. All the reasoning-graph analyses and GNN training rely solely on topological structures (e.g., edges, cliques, loops) without considering semantic content or factual verification. As a result, the proposed approach cannot distinguish between semantically correct but structurally complex reasoning and structurally similar but factually hallucinated reasoning.

**Questions:**

1. Complementary to the first point in weaknesses:
   * In line 152, the transformer layer $l$ used for extracting hidden states is not specified. Since these hidden representations form the foundation for constructing the proposed reasoning graphs, different layer choices could lead to entirely different graph structures.
   * In line 161, the node set V is defined as the set of centroids $c_k$, but in the formula at line 165, the notation $v_{i_j}$ is introduced without clear correspondence.
   * In line 180, the term "rigorous consistency-checking protocol" is said to be detailed in Appendix B, yet no concrete description or implementation details are provided.
   * In Table 2 (line 197), it is unclear why Factual-Accurate and Factual-Hallucination examples both indicate the presence of input perturbation. This seems inconsistent with their definitions.
   * In Table 3 (line 216), many symbols, such as $d(u,v)$ are neither explained nor defined. If these are intended to be elaborated in the appendix, they should not be omitted for readability and completeness.
2. Considering the limited methodological transparency, I am skeptical about the results reported in Table 4. The reasoning graphs constructed in this work are relatively small, and using such deep GNNs (with large L values) is likely to cause severe over-smoothing, making the reported performance gains questionable.

---

### Official Review · Reviewer_3xqX · 2025-11-01

**Soundness:** 2
**Presentation:** 4
**Contribution:** 3
**Rating:** 4
**Confidence:** 2

**Summary:**

This paper investigates the hallucination behavior of LLMs from a topological perspective. The authors represent a model’s chain-of-thought as a reasoning graph built from hidden-state clusters and sequential dependencies, and identify graph-structural patterns correlated with hallucinations. They propose G-Detector, a graph neural network that predicts hallucination likelihood purely from graph topology, achieving high detection accuracy (88.9% F1 / 94.1% AUROC). Using this detector, the authors further design a data-filtering pipeline that removes high-risk training samples, leading to a 13.8% improvement in factual accuracy on reasoning-intensive QA benchmarks. The work thus reframes hallucination analysis as a structural property of reasoning trajectories, rather than a purely textual or probabilistic phenomenon.

**Strengths:**

The paper shows it's possible to connect hallucination detection with graph topology. The proposed G-Detector is conceptually elegant, lightweight to train, and empirically effective at identifying hallucinated reasoning chains.
The work provides clear visualizations and statistical analyses that highlight specific graph features (e.g., path length, cycle density) correlated with hallucination.
The presentation in this paper, especially the figures, they're amazing.

**Weaknesses:**

The main concern is that: can you also compare with hallucination detection trained upon the LLM itself using the same training data, similar to a reward model? I find it interesting to know why we need GNN here, but not directly train the LLM itself (maybe with LoRa so that it's also light-weight). (seems SAR and EigenScore are prompting only)
Also for the downstream tasks the method seems only include four benchmarks (Figure 7), can you consider add some more benchmarks and cover more area, e.g. AIME, SweBench, HumanEval, WildBench... etc. for math, coding, agent, chat tasks.

**Questions:**

See Weakness, happy to raise score if addressed.

---

### Official Review · Reviewer_mDom · 2025-11-01

**Soundness:** 3
**Presentation:** 3
**Contribution:** 3
**Rating:** 6
**Confidence:** 3

**Summary:**

This paper introduces a topological framework for analyzing, detecting, and mitigating hallucinations in Large Reasoning Models. The authors conceptualize reasoning as structured graphs rather than linear text and identify 17 topological features that distinguish hallucinated from faithful reasoning trajectories. Using this insight, they build G-Detector, a graph neural network (GNN) model that detects hallucinations based solely on reasoning structure, and apply G-Detector during fine-tuning to filter structurally risky data.

**Strengths:**

1. Viewing hallucination as a topological phenomenon is conceptually novel and bridges symbolic structure with model behavior.
2. The authors analyze over 6,000 annotated reasoning graphs, systematically identifying statistically significant structural features correlated with hallucination.
3. The method provides interpretable signals, linking hallucination to “over-complication” and “thought separation” phenomena.

**Weaknesses:**

1. Constructing reasoning graphs and computing graph features for every CoT instance is expensive for large-scale datasets.
2. G-Detector is trained on the Qwen model series, which may not be generalized to other models like Llama, GPT, Claude, etc. The evaluation dataset for G-Detector is generated solely by a Qwen model. The pattern of GPT or Llama may be different. But there is no related discussion.
3. There is no discussion about the sensitivity of the G-Detector to the choice of transformer layers and graph properties.

**Questions:**

See weaknesses.

---

### Official Review · Reviewer_76eZ · 2025-11-02

**Soundness:** 3
**Presentation:** 3
**Contribution:** 3
**Rating:** 4
**Confidence:** 3

**Summary:**

This paper proposes a novel approach to detecting hallucination in long-CoT responses of LLMs. The methodology involves splitting the generated responses into segments, which are then represented in the latent space. The representations are clustered using the K-means algorithm, allowing each response to be conceptualized as a graph. By identifying specific topological features that distinguish hallucinated from non-hallucinated responses, the paper introduces a fresh perspective for analyzing hallucination in long-CoT generations. The approach offers a potential framework for training hallucination detectors and enhancing data filtering in SFT of models.

**Strengths:**

- The proposed method is both simple and intuitive, yet it proves effective in detecting hallucinations in long-CoT responses.
- The paper is well-written, with a clear structure and logical progression of ideas.
- The authors provide sufficient empirical experiments to demonstrate the practicality of the method.

**Weaknesses:**

- Some parts of the introduction to the method are not entirely clear, which may hinder readers' ability to fully grasp the underlying concepts. A more detailed explanation of key components would improve accessibility.
- Certain experimental details are omitted, which could provide further clarity on the reproducibility and robustness of the proposed method. The questions section highlights specific areas where additional information is needed.
- To enhance clarity, the superscript (l) in Equation 1 can be omitted, as its inclusion may be unnecessary and potentially confusing for readers.

**Questions:**

- Line 150-151: How are the segments split for long-CoT responses? Are there specific criteria or algorithms used to determine the segmentation, and how does this affect the detection of hallucinations?
- Line 152: How are the hidden states of each token $h_{i,t}^{(l)}$ obtained? What specific representation extractor is used in the experiments? Are the representation extractors consistent in the experiments?
- Line 400: How is the label for the test set determined? Is the test set balanced with respect to the labels (hallucinated vs. non-hallucinated responses)?

---

### Meta-Review · Area_Chair_T724 · 2026-01-06

**Summary:**

There are several concerns in the initial reviews, which remain after the discussion period.

**Reviewer Concerns:**

No rebuttals were posted.

**Reviewer Scores:**

4/6/4/2

---

### Decision · Program_Chairs · 2026-01-26

Reject